# Beyond Finding Purpose: Motivating a Translational Science of Purpose Acquisition

**DOI:** 10.3390/ijerph20126091

**Published:** 2023-06-09

**Authors:** Anthony L. Burrow

**Affiliations:** 1Department of Psychology, College of Human Ecology, Cornell University, Ithaca, NY 14853, USA; alb325@cornell.edu; 2Bronfenbrenner Center for Translational Research, Cornell University, Ithaca, NY 14853, USA

**Keywords:** purpose in life, translational science, health, wellbeing

## Abstract

A broad interest in finding purpose is understandable, as having purpose is situated in notions of “the good life” and is linked in studies to greater health and wellbeing. Yet, the empirical basis for whether purpose is truly findable is inadequate, lacking guidance from theories predicting behavioral capacities that drive its acquisition. If feeling purposeful is as favorable as studies suggest, then more transparent and precise explanations of how it is derived are needed; otherwise, the field risks illuminating this resource while leaving the pathways to it unlit. Here, I call for a translational science of purpose acquisition directed at gathering and disseminating evidence of the processes by which this sense can be cultivated. I introduce a minimal viable framework for integrating basic and applied investigations into purpose by bridging laboratory research, intervention and implementation efforts, community-engaged practices, and policies to accelerate testing and strategies for enhancing this salubrious sense in people’s lives.

## 1. Beyond Finding Purpose: Motivating a Translational Science of Purpose Acquisition

Purpose is defined as a self-organizing and prospective life aim [1]. Having a sense of purpose is a coveted asset, buttressed by enduring philosophical treatments as a virtue and resource for fulfillment and enjoyment of “the good life”. Across varied national and cultural contexts, polling data suggest high levels of public interest in purpose and the sources that contribute to it [2,3]. This interest runs parallel to a groundswell of recent academic investigations into purpose. Within this growing research corpus there is substantial empirical evidence linking sense of purpose with important indicators of health and wellbeing, such as greater happiness [4], greater resilience to stress [5], more social connections [6], greater cognitive function and mental health [7,8], greater income and net worth [9], physical health and functioning [10,11], and lower risk of morbidity and mortality [12,13,14]. This generous array of correlates has emerged in studies conducted from early adolescence to older adulthood [15,16], lending credence to the role of purpose in human flourishing and as a psychological asset well worth having.

Thus, genuine interest in how one acquires purpose is understandable. A prominent colloquial view is that purpose can be found by individuals who seek it [17]. Indeed, a litany of books and self-help guides, massive online courses and trainings, and career development resources prescribe purpose as something to find [18,19,20]. Though often publicly promoted, the idea that purpose can be expressly found has curiously scant scientific backing. Operating perhaps more as metaphor than a precise mechanism, no prevailing theories of purpose claim *finding* as the action by which a person who does not have purpose ultimately acquires it. It is worth noting that much of the available research literature on purpose is comprised of correlational effects—demonstrations of how much variance in health and wellbeing is explained by individual differences in sensing purpose. Far less research has focused on the behavioral capacities that give rise to sensing purpose in the first place. Stated critically, efforts to convince people to find purpose rely on inadequate or remarkably opaque evidence for explaining how they ought do so. Purpose researchers have yet to conclude precisely which capacities (e.g., neurological, physical, cognitive, affective, social, etc.), functioning to what degree, and under which ecological conditions, are necessary to find purpose. Without an empirical foothold on the specific processes that underlie purpose acquisition, nudges to find it may be misguided, flawed, or even harmful [21,22]. Avoiding these pitfalls may require greater efforts to infuse lay perspectives on purpose with scientific evidence.

In the absence of formal theories charting how purpose is acquired, a scientifically rigorous foundation is needed to (a) establish theoretically derived predictions about behavioral capacities that give rise to measurable gains in purpose, (b) conduct ecologically valid tests of these claims, and (c) implement purpose-enhancing practices through evidence-based interventions and programs. Establishing such a foundation is challenging as it requires field-organizing efforts to calibrate and accelerate scientific inquiry and its dissemination across multiple levels of stakeholders. Yet, a serious interest in doing so could overcome these complexities by leveraging advances in translational scientific approaches, which support systematically integrating basic research with real-world application to enhance desired outcomes and address urgent needs [23,24]. In the broader field of purpose research, this effort would be transformative. For example, a current PsychInfo search for all texts published between the years 1983–2023 including both the subjects “purpose in life” and “translational” returns fewer than 10 articles in total. The timeliness of translating research on purpose is difficult to overstate, given the landscape of formidable challenges facing individuals’ quality of mental health, workplace engagement, and sense of mattering in society [25,26,27]. If purpose corresponds with wellbeing to the extent suggested by research, then identifying and sharing sound methods for acquiring more of it surely qualify as urgent, underscoring the present appeal for a translational science of purpose acquisition (hereinafter TSPA).

In motivating a TSPA, I first discuss features of the purpose construct that make it suitable for translation. Next, I introduce a minimal viable framework to guide the collaborative knowledge exchange between researchers and community partners co-invested in identifying the behavioral capacities needed for purpose acquisition. I also address innovations that could enhance the minimal viable framework and further hasten strategies for promoting those capacities deemed promising for broader constituencies. I close with a consideration of the consequences of pursuing a TSPA for fostering healthy populations, with an emphasis on marginalized communities for whom the benefits of traditional scientific efforts are too often the least ensured. Setting the stage for this agenda, I offer a brief primer on translational science.

## 2. Translational Science: A Primer

I wish to avoid a confusion that can arise when the terms translational research and translational science are viewed as synonyms. Translational research refers to investigative methods that bridge basic or laboratory discoveries to practical applications and policies that benefit people, whereas translational science refers to the study of those methods themselves in order understand and improve the predictability and efficiency of knowledge exchange [23,28,29]. Of the two terms, translational science is the broader and more self-reflective, aimed at uncovering insights into effective analytic approaches, implementation methods, and operational principles that achieve the desired aims of using research to help people. Translational science can be recursively thought of as the science of translation—responsive to how the search for basic understandings can be optimally integrated with the search for applied solutions. To be sure, exemplary translational research projects are already underway in the literature on purpose [30,31], showcasing the value of conducting studies with the goal of improving lives. However, these examples are discrete and lack coordinated integration with one another such that learnings in one project have no immediate relevance for the next. By spotlighting translational science here, I intend to motivate field-level considerations to achieve more integrative aims and make purpose acquisition a more predictable and efficient endeavor.

Of course, drawing too sharp of a distinction between basic and applied work may reify longstanding and arguably fictional divisions of what some view as a spectrum of scientific inquiry [32]. However, by acknowledging different motivations that may drive efforts along that spectrum, the benefits of naming translation as an overarching aim of the scientific enterprise come into view. First, the pace by which basic research findings make their way into real-world use is slow, with some estimates suggesting a lag of a decade or more [33]. Translational science is equipped to address this issue by detecting inefficiencies and promoting alternative processes that hasten knowledge exchange. Second, translational science can accelerate system-wide learning by capitalizing on bidirectional feedback loops between investigators and community partners in one place to share insights with those operating in others. Rich examples of this kind of integrated learning have been documented in biomedical and allied health fields wherein problems that arise in one locality can be avoided elsewhere, substantially lowering costs and burdens to researchers and participants and promoting more regenerative practices [34].

Third, translational science can fill important gaps left unaddressed between sectors along the scientific spectrum. For instance, because every phase of the translational process needs funding and resources to operate, no single sector should be overburdened by ensuring the adequacy of resources for all others [23]. Toward addressing these gaps, translational science agendas have been shown to catalyze the creation of new structures (e.g., funding agencies or novel grant mechanisms within agencies) and establish priorities (e.g., setting goals to improve communication across translational efforts or to increasing community participatory practices), which can solve systemic problems that neither researchers, community stakeholders, nor policymakers could resolve alone.

The utility of translational science is found in its ability to help identify and remove barriers to success and enable measurable and enduring improvements in people’s lives. When successful, translational science produces results that can inform the design and implementation of policies and organizational practices that support positive outcomes within specific domains. Because translational science is optimized when multiple perspectives are included and intentionally integrated [35], it can invite more engagement and participatory strategies than mere laboratory research has achieved. This point is important to the extent that science—and those who conduct it—are understood to be at their best when open, transparent, and promoting well-being and social progress for all [36].

## 3. Purpose Inquiry and Its Suitability for a Translational Science Approach

Some observers may be unconvinced that extant research on purpose is poised for scientific translation. For example, there is currently no unifying definition of purpose that all researchers employ. Even among the available definitions, many suffer vague boundaries that allow conflation between purpose and concepts such as meaning (which typically involves coherence and mattering) and goals (which typically refers to concrete and achievable objectives) [37,38,39], while others present conflicting structural arrangements with purpose sometimes depicted as a subcomponent of meaning [40], and other times meaning depicted as a subcomponent of purpose [41]. These discrepant characterizations then inform the development of purpose measures that each emphasize different aspects, complicating efforts to generalize findings. In addition, the literature on purpose has long tolerated bifurcated lines of inquiry of the same overarching concept: with one branch focusing on sense of purpose—the perceptible feeling that one has overarching life aims and worthwhile engagements, and another focusing on purpose in life—the specific content of one’s ultimate aspirations. Though the confusions caused by this divergence have been elaborated elsewhere [42,43], it is worth noting that associations between purpose and indicators health, wellbeing, and positive adjustment are almost exclusively documented within studies assessing sense of purpose, and not the disclosure of its content. To the extent this issue presses on issues of finding purpose, a question arises: which form of purpose should individuals find—a greater sense of it or the capacity to articulate its content?

These challenges, however, are not unique to the study of purpose and are not detrimental for establishing a meaningful translational science agenda. In fact, definitional and measurement complexities surrounding constructs can offer useful points of entry for involving broader stakeholders in shaping research questions and study designs relevant to the issues they are facing [44]. Beyond the recognized challenges, three features of contemporary purpose inquiry are worth discussing as they afford useful starting points in constructing a TPSA. These are (a) understanding that the experience of purpose requires multiple disciplinary perspectives, (b) viewing volitional exploration and commitment processes as a conceptual basis for developing more precise theories of acquisition, and (c) understanding linkages between sense of purpose and biological mechanisms as opportunities to validate gains in purpose over time. Each of these areas are elaborated below.

### 3.1. Purpose as a Biopsychosocial Experience

First, the literature on purpose supports viewing it through a biopsychosocial perspective [15]. That is, understanding what purpose is and determining its beneficial role in people’s lives requires considering a complex interplay between biology, psychology, and social contexts. From a biological perspective, sensing purpose is rooted in brain networks, as well as in the physiological systems that support cognition, emotion, and behavior. From a psychological perspective, purpose is shaped by individual characteristics, such as personality traits and cognitive abilities, each of which supports specific explorative and commitment capacities. Finally, from a social perspective, purpose is influenced by the broader systems in which individuals are embedded, including family, school, work, and community contexts. By acknowledging this multifaceted nature of purpose, determining factors that could plausibly give rise to it—or contribute to reliable gains over time—require a paradigm that can bridge these different disciplinary levels of analysis.

### 3.2. Exploration and Commitment as Foundational Elements of Purpose Acquisition

Second, key parts of the literature on purpose have been constructed in proximity to research on identity, as both are thought to entail self-exploration and discovery. According to influential perspectives on identity [45,46,47], people construct a sense of identity by exploring plausible self-definitions that lead to commitments over time. Thus, identity exploration and commitment are developmental processes that work together to shape an individual’s sense of who they are. Likewise, exploring and committing to a sense of where one is heading are believed to represent related but distinct processes that underlie the formation of purpose [48,49]. Both conceptual frameworks [50] and recent field studies [30,31] affirm that when people are provided with opportunities to consider, practice, and reflect upon purposeful experiences, they exhibit gains in both purpose exploration and commitment. Combined, exploration and commitment processes provide an initial basis for viewing purpose as a sense that is not merely found, but as one that is volitionally cultivated via active engagement in activities afforded by one’s surroundings.

However, which specific behavioral capacities aid purpose cultivation? Cognitive capacities that enable reflection, memory, sustained attention to ongoing activities, as well as prospection and planning have been implicated in sensing purpose [8,30,51]. The ability to perceive one’s life as not simply a moment but as a pathway connecting past activities to future outcomes appears to be important for forming a sense of purpose, as does the ability to be physically active and behave in ways commensurate with available opportunities in one’s environment. Studies show that physical activity [52], especially daily activities deemed meaningful to an individual [53], contribute to gains in sense of purpose. Importantly, these capacities and the environments that support them are not distributed equally across people. Sumner et al. [54] detail a constellation of barriers or challenges some might encounter when trying to cultivate purpose against the backdrop of physical or social constraints that may inadvertently or intentionally undermine them. Thus, theories of exploration and commitment processes thought to be conducive to purpose acquisition can be more precisely articulated by incorporating details about the functional capacities of individuals and the contextual affordances that support their pursuit.

### 3.3. Biological Mechanisms Linked to Sense of Purpose

A third indicator that purpose research is poised for scientific translation is the emergence of data linking sense of purpose with biological mechanisms. In the past two decades, there has been substantial growth in assessments of purpose alongside tests of neurological functioning, physiological regulation, and physical health [14,27]. Neural imaging studies, for example, have demonstrated that individuals who report a greater sense of purpose exhibit greater connectivity in the brain networks implicated in self-monitoring and goal-processing [55,56], and those related to decision-making [57]. In addition, sense of purpose has been linked with greater overall physical health, marked by consistent patterns of lower instances of chronic disease, stress reactivity, and overall allostatic load [58,59,60]. Taken together, these biological indicators provide a generalizable foundation atop which more phenomenological experience of purpose can be measured, explained, and translated.

Uncovering biological mechanisms is an important development for a field wishing to translate its discoveries. Doing so can enhance the sophistication of researchers’ ability to detect and differentiate between sources of effects commonly observed, and design interventions better tailored to the needs of specific populations. As applied to the concept of purpose, disagreements about definitional qualities can be more easily reconciled by establishing the biobehavioral systems one expects to be activated by the presence of purpose. Likewise, measurable gains in purpose should be expected to have resonance within those biobehavioral systems, lessening the need to adjudicate between competing self-report measures. While a fuller rendering of all mechanisms tethered to sensing purpose exceeds the current scope, studies actively exploring these mechanisms present exciting areas of growth that connect it to other fields such neuroscience and biology. However, they also tie purpose inquiry to even broader disciplinary foci, such as sociological and cultural studies, insofar that questions of whether a pursuit of purpose is primarily an individual or collective exercise, or whether it will be valued by other members of one’s community or society, await answers.

## 4. Introducing the TSPA: A Minimal Viable Framework

Understanding purpose as a psychological resource that can be cultivated, a translational science framework capable of explaining how to do so must integrate multiple levels of inquiry, analysis, implementation, and dissemination. Figure 1 presents a minimal viable framework that could guide efforts by sequencing specific translational research tasks (inner most triangles) that tether core scientific priorities (outer most quadrants), and that are paved by constant touchpoints by which community stakeholders can inform the research process (intermediate ring). Of course, this minimal viable framework will benefit from tailoring and elaboration by the stakeholders who use it. The intention is not to construct a conceptual model too simplistic to be of use, but rather to offer some foundational scaffolding that can be responsive to the needs of those invested in it. To examine the utility of this scaffolding, I discuss how its components might address questions of purpose acquisition.

At the broadest level, a TPSA should seek to integrate four core scientific principles that guide construct operationalization, methods for the collection and analysis of data, design and implementation of interventions, and the construction and delivery of evidence-backed policies. Initial determinations of what purpose is and how it is to be measured should be based on understandings that have both theoretical and practical resonance, satisfying calls for greater verisimilitude and plausibility in psychological theories [61]. As an illustrative example, Burrow et al. [31] partnered with educators working in a statewide youth development program called 4-H to determine if youth participation gave rise to a sense of purpose among enrolled youth. The overall project began with researcher-led interviews about how educators understood the concept of purpose, and which features of 4-H programs educators believed the most likely to facilitate gains in youth purpose. The results of these interviews helped refine a survey that was administered to youth, asking about which program features they had engaged in and to what degree. Inferences gleaned from those surveys could be used to make changes in overall program delivery as deemed appropriate by 4-H educators. Throughout the process, investigators and educators engaged collaboratively in workshops, trainings, focus groups, conference presentations, and even publications [62,63], enabling bidirectional feedback leading to a conclusion about how 4-H cultivates youth purpose. Insights can be scaled across county- and state-based programs while retaining the ability to make local adjustments.

Connecting the four core principles are the specific translational research activities that can produce the empirical foundation for promoting purpose acquisition. These activities might begin with researchers building from known strategies for prompting people to consider purpose in their lives. For instance, a rich literature on wise interventions [64] suggests that even brief opportunities to consider or reflect upon one’s purpose can give rise to momentary upticks in sense of purpose [65], resulting in benefits resembling those experienced by individuals reporting high levels of dispositional purpose [66,67]. Insights from these interventions may point to vital capacities needed to reflect on purpose routinely that give rise to a more enduring sense of purpose over time [50]. That is, actually tracking sense of purpose in real-time, and documenting the contexts and circumstances present when such feelings arise may provide a rich collaboration opportunity for individuals and researchers to learn how to promote more of it. No different than a cardiologist tracking heart rhythms with a wearable monitor, while a patient charts the contexts they are in when they feel palpitations, sense of purpose could be tracked in similar ways. If such monitoring was determined successful at identifying and maintaining a sense of purpose, researchers and community stakeholders may work with policymakers to extend and implement more systemic opportunities to help others who wish to feel more purpose in different settings. That is, integrating empirical evidence with practical guidance could inform interventions that help people acquire a sense of purpose can lead to the development of evidence-informed policies more broadly. Indeed, similar translational trajectories have emerged with respect to leveraging the science of adolescent sleep on academic performance, with real impact on policies regarding school start times [68].

As depicted in the minimal viable framework, translational science requires sustained engagement and communication between researchers and community stakeholders. Thus, critical areas a TPSA might investigate are those that bolster efficiencies in knowledge sharing, community-engaged research, and dissemination and outreach practices aimed at understanding and facilitating experiences that help people feel more purposeful. In support, the National Institutes of Health has established the National Center for Advancing Translational Sciences [24] dedicated to enhancing capacity and optimizing translational scientific efforts and infrastructures. Likewise, other funding agencies and translational science centers have designated supports and training opportunities to help researchers learn and demonstrate sound translational practices, including integrating community-based participatory practices [69]. Ultimately, community members are not passive recipients of scientific pursuits, nor are researchers mere laboratory operatives without connection to a community. These groups are not mutually exclusive and need each other for ultimate success in the translational science agenda.

Offered here merely as a starting point, the minimal viable framework for a TSPA could be enhanced by infusing transformative ideas for improving public science [70,71]. Among such ideas, scientists can build inclusive research teams with diverse perspectives and experiences that enable meaningful connection and engagement with community partners and demonstrate these priorities through training in translational approaches in graduate programs [36,72]. Other innovations may involve remaining open to entering the translational cycle at different spots along the way. There are numerous examples of communities already engaged in practices thought to imbue people with a sense of direction and life engagement [73,74]. Researchers’ willingness to lend their investigative skills to evaluate the effectiveness of existing practices that may serve purpose acquisition may be an important and validating point of entry to a translational science system. Moreover, relying on systematic reviews and meta-analyses wherein the sample characteristics align with the communities engaged afford a more advantageous basis for designing studies and interventions than presume all projects must begin with an original laboratory study. Bringing the full toolkit of scientific inquiry not just to project design and interpretation, but to innovation and systemic improvement of the overall inquiry process, is at the heart of the translational science agenda.

## 5. Conclusions

Embarking on a TSPA affords opportunities to put the science of purpose to timely use. Importantly, the present call for a TSPA should not be interpreted to in any way stifle laboratory or basic investigations into purpose. In fact, steady investment in basic research can fuel more content for translation and upstream impact. However, the overall translational science agenda may show its greatest promise when considering the communities most in need of engagement and directing efforts toward those first. For example, calls have been made for greater research into opportunities that enhance sense of purpose among members of marginalized communities for whom social structures may extinguish opportunities to pursue interests most aligned with their identities and interests [54]. In this way, pursuing a TSPA could align with recent calls for grounding scientific practices with the intention to restore and heal those who have experienced marginalization or harm [71], sometimes at the hands of scientific pursuits themselves [75].

As such, there is one final consideration to note, not as an ancillary point but as a critical accent wall against which my central thesis can be contemplated. Constructing a TPSA presents an opportunity to direct a scientific agenda toward greater social justice and equity. The demonstrated association between purpose and not only how long we live but with how well we live makes knowing how to acquire this sense worth substantial scientific investment. That purpose corresponds with greater contribution and generativity [76], volunteerism [77], and commitment to making social and global impacts [78], means that efforts to increase sense of purpose will pay dividends for the individuals who possess it as well as the collective. The health and wellbeing of populations stand to be improved by understanding how to acquire purpose, which could lower healthcare costs and caregiving burdens downstream [79]. If a sense of purpose is not simply found but is instead actively and intentionally cultivated, using science to help people become purpose cultivators promises to serve us all well.

## Figures and Tables

**Figure 1 ijerph-20-06091-f001:**
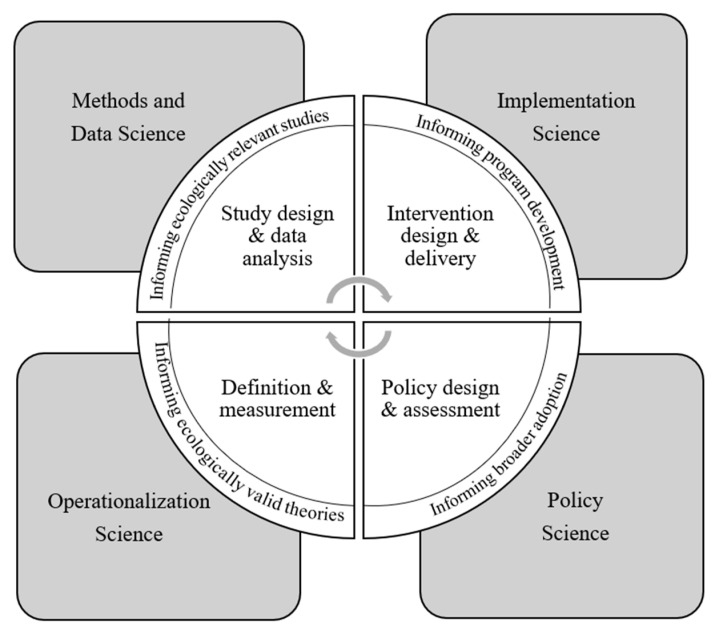
A minimal viable frame for a translational science of purpose acquisition. Note: the four quadrants represent core scientific priorities that underlie the translational process. The inner-most components represent the research and practice activities undertaken in translational research, which can be observed and improved upon by the broader translational science priorities. Surrounding these activities are ways in which community stakeholders can inform the entire translational process.

## Data Availability

No new data were created or analyzed in this study. Data sharing is not applicable to this article.

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
