# Peer review of "Beyond Finding Purpose: Motivating a Translational Science of Purpose Acquisition"

_ijerph, 2023, doi:10.3390/ijerph20126091_

Round 1

Reviewer 1 Report

The theme of this manuscript is interesting and is developed in a coherent and logical manner. However, in my view, it is closer to an opinion article or academic reflection than to a scientific review article, as the review itself lacks depth.

In other words, constructing a TPSA for studies on purpose in life may seem logical or necessary, but in scientific terms a factual justification is needed, based on the analysis of shortcomings seen in current studies and problems highlighted by authors of great relevance in the field. While the latter can be said to be fulfilled, the former point is nowhere to be found in the manuscript. 

Furthermore, the analysis of the shortcomings of current scientific works must be carried out with rigour and depth, following a systematic method that may or may not be exhaustive, but which shows the reality of the object of study from which the contributions are subsequently made, which is the part that is represented in the text.

For these reasons, I believe that this project needs a genuine scientific justification, based on an in-depth scientific review of current affairs, which would entail such a profound change to the manuscript that it is not publishable in this state, in my opinion, but which could be included in another type of space, more oriented towards academic reflection.

On the other hand, it should be noted as a minor shortcoming that the bibliography lacks the necessary web links, most of the DOIs are missing and many of the references need to be reviewed and corrected, in addition to the fact that the MDPI citation style format has not been used in the manuscript.

Author Response

Though this reviewer did not favor publication, I want to express my gratitude for the time they took to review my manuscript and for the comments they provided. I do appreciate that they rated the paper highly on comprehension and organization. 

As I understand it, this reviewer does not think the manuscript warrants publication because it lacks a scientific justification or adequate rigor. Ultimately, the aim of this paper is to argue that the field of purpose lacks theoretical foundation on which to study - and communicate its findings to the public - precisely how people develop and maintain a sense of purpose. I situate this argument squarely within the void of existing citable frameworks that currently achieve the aims I just stated. While the lack of translational theories to review certainly truncated my ability to unpack them, I continue to believe this paper adds substantially to the scientific literature on purpose as a point of departure for constructing a translational science of purpose acquisition. Moreover, I submitted this paper as a review - and at ~80 references included in a 4,000 word manuscript, I believe I have included a significant review of relevant works.

The one aspect they felt could be addressed is correcting the reference list to be commensurate with journal standards. I have done so in my revised paper. 

Reviewer 2 Report

I believe that submitted manuscript is a good article that scientifically unravels the way to acquire purpose, purposeful feeling, or sense of purpose. I enjoyed reading the manuscript and there were many things I could agree with. If only a few things related to manuscript writing are supplemented, it is evaluated to be at a level that can be published in Q1 journals. What I think needs to be improved is:

1. Please correct the citation and reference because this journal does not follow the APA citation and reference style.

2. In the abstract, you mentioned that finding purpose wold be positive for an individual's health or well-being, but it would be better if this was stated convincingly at the beginning of the introduction. Because there could be side effects of pursuing a purpose too much or negative aspects of the sense of purpose.

3. It would be better to present Final Considerations and Consequences of Pursuing a TSPA in more detail.

Author Response

I am grateful for the comments and suggestions provided by this reviewer. I have made the following changes in response to the feedback I received.

  1. I have corrected the citation style and reference list, from APA to AMA.

  2. I appreciated this comment. As suggested in the Abstract, I believe the current Introduction clearly suggests sensing purpose in life corresponds with greater health - with multiple indications of such linkages provided. That said, I have added one more very recent citation for an empirical study linking sense of purpose with greater physical health.
    1. While I suppose I agree there could be side-effects of pursuing one's purpose, I am unfamiliar with empirical studies documenting such effects. The closest I can reason are studies of John Henryism - in which expending too much effort to achieve as a stress-coping strategy can overrun allostatic load. But even still, recent studies suggest greater sense of purpose correspond with lower allostatic load. 
  3. Space / Word-count was a limitation in this manuscript, so I did not describe Final Considerations in extensive detail - though each consideration is largely drawn from points drawn out in prior sections. However, I have now added more detail in a key section of  that section. 

Thank you for this very helpful review. Attempting to reconcile these points has led to a much improved manuscript.

Reviewer 3 Report

This review article is a call for a co-ordinated effort towards the study of Purpose Acquisition. It provides a balanced presentation of translational science, research of purpose, and whether the the latter is ready for the former. A theoretical structure is proposed regarding the development of a translational science of purpose acquisition and the necessity of communication between these components in order to more efficiently achieve these goals is described.

This paper is excellent. I don't think requires any changes.

Author Response

I thank this reviewer for taking the time to read my manuscript, and for suggesting that I succeeded in what I set out to accomplish with it. I am especially pleased they found no aspect in need of improvement

With gratitude,

-author

Reviewer 4 Report

Good day, I have reviewed your paper and, I have to say that it is written very well for the purposes it tries to achieve. There are however some suggestions that I think will make the article more accessible and help serve its purpose better.  1. Regarding the distinction between meaning and purpose, this should be made more clear, with at least a proper definition of purpose, if not meaning as well.  2. From the standpoint of a scientific paper, this article reads different, being more in the realm of non-fiction literature (or an editorial), which is good, as it makes it more accessible to readers (which, as you stated in the paper, is important, as researchers are part of the community, and a better informed and educated community is a path toward increased equity), but the paper is made harder to read by two factors.  A. Firstly, though the article is written masterfully, the niche for readers of such sophisticated English is quite limited. Simpler phrases with shorter and more common words would serve the paper better.  B. Secondly, there are many times in which a call to action is proposed, asking for specific acts (such as research concerning the elements that give One's life purpose, rather than the effects of the subjective feeling of purpose), but being rather vague in the ask. I think that (at least an example of the proposed) specific recommendations could help guide future purpose researchers.  The paper is both too long and too short in this way, spending precious lines and paragraphs trying to make the case for TSPA, but also providing little more than vague guidelines (with the exception of the mininum viable framework, but that could also be made a little more specific) for its proper implementation.

Author Response

I thank the reviewer for this set of comments. I believe addressing them has been helpful for constructing a stronger paper. Below, I describe is how I've addressed each suggestion this reviewer raised. *Numbers correspond to the reviewer's points.

  1. I now more clearly distinguish purpose from the concept of meaning. See page 3. 
  2. Throughout the manuscript, I have tried to shorten sentences where I felt doing so could prove helpful without detracting from the larger point. I hope these subtle changes add up to a paper that is less 'sophisticated' in its wordage and easier to read overall.

Moreover, I have added a much more precise call to action concerning the need to study how people cultivate a sense of purpose (as opposed to any particular purpose content). See page 7. 

Again, I thank this reviewer for their thoughtful suggestions.

  1.  

Round 2

Reviewer 1 Report

The improvements do not represent a profound change to what was previously assessed, so my opinion remains unchanged. I am sorry I cannot change my opinion.

Author Response

I once again thank this reviewer for offering to read the revised version of this manuscript. Though their opinion has remained unchanged, I have made two additional revisions to the manuscript that I believe improve it to be more in line with their original concerns. I then offer one additional justification / explanation for why I've constructed the paper this way.

Two changes made in the current revision:

  1. On page 1 (likes 41-43) I now raise the point that there are no theories of purpose that offer a mechanistic account of 'finding' as the process by which purpose is acquired. In doing so, I contrast this lack of scientific argument with the common usage of 'finding' in lay perspectives on purpose (with cited examples given earlier in the paragraph).  Here, I am making the argument that lay perspectives on how people derive purpose could be informed by scientific evidence -- if only there were formal theories of precisely how people actually derive an enduring sense of purpose. I then proceed to describe how such a theory could be constructed via translational methods.  *As I stated in a previous response, the reviewer may not agree that this claim is a worthwhile scientific argument. But I believe that it is - and hence spend the rest of the paper contending how the field might go about it and what is to be gained if it did so.
  2. A second addition is the insertion of a PsychInfo search considering the terms "purpose in life" and "translational" simultaneously. The results of this search returned a total of 8 articles that used these terms in the same publication (thought I note, many of those only used the term "translational" in the reference list of a given publication. In any case, I inserted this point to further bolster my claim that a translational science approach to purpose acquisition would contribute to the existing literature. *I note the reviewer rated this manuscript as 2(of 5) on its potential to contribute significantly to the field. To me, this PsychInfo search suggests it has the potential to add substantially to a field that has largely omitted the insights within - just as many other fields have benefitted when similar calls were made.  Of course, I humbly disagree with this reviewer on this rating, though I suppose all authors believe their work stands to contribute to a field. What is perhaps most agreeable though is that this work cannot contribute to the field if it is unpublished. 

One final point I wish to raise is that it seemed to me from the reviewer's first round of comments that the issue here is less about the the content relating to purpose itself and more about whether the format and style of writing are appropriate for a review paper in this outlet.  That is, if this were to be considered as an "academic reflection" or perspective or commentary then maybe it would be more appreciated.  *I certainly cannot speak for this reviewer, but that is my understanding of the issue. But no other reviewer noted this concern.

Ultimately, I leave it to the editor to decide if my submission fits with the aims of the journal. But as for the 'scientific soundness' of this article, I was responding to a call for the special issue on the science of purpose and wellbeing. I submitted an abstract detailing what I would cover and remained true to those aims in doing so. My abstract was invited for full submission. In it,  I am raising a call for translational science in this area and review research that could help those interested in doing so accomplish this goal.

Perhaps the editor could adjudicate which specific additions would be needed from here.

Again, I am grateful for the time this reviewer has spent considering this manuscript and my responses.